# Liquid Oil Trapped inside PVA Electrospun Microcapsules

**DOI:** 10.3390/polym14235242

**Published:** 2022-12-01

**Authors:** David Mínguez-García, Noel Breve, Lucía Capablanca, Marilés Bonet-Aracil, Pablo Díaz-García, Jaime Gisbert-Payá

**Affiliations:** 1Departamento de Ingeniería Textil y Papelera, Universitat Politècnica de València, 03801 Alcoy, Spain; 2Centre for Textile Science and Engineering, Univeristeit Gent, 9000 Gent, Belgium

**Keywords:** shell, essential oil, sage, thyme, nanofibers

## Abstract

Electrospinning makes it possible to obtain solid fibers, in addition to core-shell fibers, using coextrusion. However, an exhaustive control of parameters allows the core-shell fibers from emulsion electrospinning to be obtained. The solvent in the outer surface tends to evaporate and the polymer density increases, moving the emulsion drops towards the center, which in turn promotes coalescence, thus creating the core. The aim of this work was to avoid coalescence and obtain a net of nanofibers entrapping oil microcapsules. We obtained an emulsion oil in water (O/W), with polyvinyl alcohol (W) and two essential oils (O), sage and thyme. An electrospinning process was used to place the microcapsules of oil inside a net of nanofibers. The electrospun veil was characterized by organoleptic testing, SEM microscopy, FTIR spectroscopy, DSC thermal analysis, and pressure tests. Organoleptic testing, FTIR spectroscopy, and DSC thermal analysis demonstrated the presence of the oil, which was retained in the spheres observed by SEM microscopy, while pressure tests revealed that the oil remained in a liquid state. Furthermore, we demonstrated a strong relationship between the emulsion size and the final microcapsules created, which are slightly larger due to the shell formation. The size of the emulsion determines whether the spheres will be independent or embedded in the nanofibers. Furthermore, the nanofiber diameter was considerably reduced compared to the nanofibers without the oil.

## 1. Introduction

NF veils have a wide range of applications. They can be used for filtration [1,2], biomedicine [3,4,5], or even for protection against COVID-19 [6,7,8]. NFs can be made of different polymers, such as polypropylene (PP) [9], polyamide (PA) [10], polyvinyl alcohol (PVA) [11], polylactic acid (PLA) [12], etc.

Electrospinning makes it possible to obtain solid fibers, in addition to core-shell fibers, using coextrusion. When the core is not fed, hollow nanofibers are produced. The production of nanofibers through electrospinning is controlled by a multitude of parameters that affect the final result. These are solution parameters: polymer concentration, viscosity, surface tension, molecular weight, conductivity, and solvent volatility; variables of the electrospinning process: voltage, supply flow, type of collecting surface, and distance between electrodes; and environmental parameters: humidity, temperature, and air pressure. However, in recent years there has been an increase in the number of publications related to sustainable electrospinning, or green electrospinning. This type of electrospinning is mainly characterized by the solvent used, which can be the universal solvent, water, or organic solvents, the latter not meaning those that can be classified as green electrospinning [13,14].

The encapsulation of active principles, essential oils, drugs, enzymes, vitamins, etc., inside the nanofibers has led to great advances in different industrial sectors, such as in filtration, in the controlled release of drugs, in dressings for wounds, and in the immobilization of enzymes. Taking this to the next stage, it is possible to electrospin the active compound dispersed within the polymer, i.e., an emulsion.

Emulsion electrospinning has recently become one of the techniques used to obtain core-shell fibers without the need for the coaxial electrospinning system. In emulsion electrospinning, the starting point is an emulsion which can be O/W (an oil is emulsified in an aqueous solution), in order that, due to the conditions of the emulsion and the electrostatic field, the polymer solidifies. The solidification of the polymer increases its density, displacing the emulsified oil droplets towards the center of the polymer jet and, therefore, the oil tends to be located in a central channel, after which coalescence of the droplets occurs. As a result of the fusion of these droplets, the core is finally generated and the polymer from the aqueous phase surrounds it, forming the shell [13,15,16]. During the electrospinning process, the solvent in the near-surface region evaporates faster than the central part of the polymer, leading to a rapid increase in the viscosity of the outer layer compared to that of the inner layer. Consequently, movement of the droplets towards the inside of the polymer jet is induced, and the oil droplets are simultaneously condensed and stretched into elliptical shapes in the axial direction of nanofibers under the force of a high-voltage electric field [15]. The high-velocity jet is also subject to drag forces derived from its interactions with the surrounding air, which can play a key role in the deformation and attenuation of the emulsion droplets. Moreover, other forces, including gravity, the repulsive force between two charges known as Coulomb’s law, surface tension, and viscoelastic forces also act on the charged jet [17]. Repulsion according to Coulomb’s law and electrostatic forces are responsible for the expansion of the droplets. Surface tension and viscoelastic forces contribute to the contraction of the charged droplet to minimize the interface area between the air and the jet [18].

There are many types of oils, but essential oils are characterized by their extremely diverse range of applications in the fields of medicine [19], biology [20], and aromatherapy [21], among others. Their stability has been extensively studied, but there is a broad consensus that these oils are highly sensitive to light, temperature, and oxygen, among other factors [22]. Electrospinning is a protection mechanism for these oils, allowing their gradual release.

In this work, we focused on the encapsulation of two essential oils: sage (S) and thyme (T), by means of the technique of emulsion electrospinning, from a solution of polyvinyl alcohol (PVA). The preparation of stable O/W emulsions requires exhaustive control of process parameters such as the type of agitation, speed, time, temperature, oil/water proportions, presence or absence of surfactant, etc. When these emulsions are intended for electrospinning, it is extremely important to keep the emulsion stable for the period elapsed between the preparation of the emulsion and the application of the electrospinning process. PVA has been selected as a polymer because it is a water-soluble polymer and is compatible with electrospinning. Thyme has been chosen as an essential oil because it has antioxidant and antimicrobial properties [23] and contains thymol, one of its components and one of the representative elements of essential oils [22]. The essential oil sage has also been selected because, like thyme, it has antioxidant and antimicrobial properties [24]. The objective of this paper was to demonstrate the possibility of generating PVA microencapsulated oil from emulsion electrospinning. The authors aimed to demonstrate the presence of the oil within the PVA capsules, characterize the capsules’ shapes and sizes, determine the ideal electrospinning conditions, and test whether the encapsulation maintains the oil’s antibacterial properties.

## 2. Materials and Methods

### 2.1. Materials

Polyvinyl alcohol (PVA) Mw 61,000 g/mol, purchased from Sigma–Aldrich (Akralab, Alicante, Spain), was used to obtain the emulsions. The solutions were prepared with distilled water. The essential oils purchased were sage and thyme, both purchased from Lozano Essences (Esencias Lozano, S.L., Murcia, Spain). Tween 80 from Panreac (Akralab, Alicante, Spain) was used as surfactant.

### 2.2. Methods

To obtain the emulsions, a 9% (*w*/*v*) PVA solution was prepared by heating the water to 80 °C with magnetic stirring until the PVA was completely dissolved. Subsequently, 2% or 4% (*v*/*v*) of the essential oil to be tested was added to the PVA solution. The essential oil was added to the PVA at room temperature. Initially, the PVA solution was placed in the reactor and the homogenizer was activated at the revolutions established for each test. When the desired revolutions were reached, the amount of oil was gradually added. Once the addition of the oil was complete, the revolutions were maintained for a period of 60 or 3 min. The homogenizers and conditions were as follow: a propeller homogenizer (500 rpm or 1000 rpm; 60 min) or Ultraturrax homogenizer, and a toothed accessory (7000 rpm; 3 min).

The emulsions obtained were characterized using different methods. The viscosity of the emulsions was measured with the Visco Elite R viscosimeter. The measuring device was selected according to the manufacturer’s instructions for the measuring range obtained. For each of the emulsions, in addition to the viscosity, the conductivity was measured with a Crison Conductimeter Basic 30 (Hach Lange Spain, S.L.U., L’ Hospitalet de Llobregat, Spain). The surface tension was measured as well, using a Krüss tensiometer K9 (Krüss, Hamburg, Germany).

The electrospinning process was carried out with a BIOINICIA electrospinning system (Bioinicia, Paterna, Spain). During the electrospinning, the nozzle-collector distance was 15 cm. The collector was placed vertically. The same flow rate (0.5 mL/h) and voltage values of 14 and 20 kV were used. Subsequently, after adjusting the voltage, electrospinning was carried out for periods of 15 min.

The first characterization carried out on the electrospun samples was the detection of the essential oil by means of an organoleptic test. To do this, the samples were presented to 5 individuals, and they were asked to identify odor in front of samples referenced with letters of the alphabet, avoiding the power of suggestion on the volunteers. Both the fabrics with PVA nanofibers and those with the essential oil in the nanofibers were shown. They were asked to identify whether or not they smell any essence and then, if possible, to identify the type of oil.

For the morphological characterization, FESEM ULTRA 55, (Carl Zeiss, Jena, Germany) scanning electron microscopy (SEM) was used, using an accelerating voltage of 2 kV on the surfaces to be analyzed of each of the samples and at the magnifications considered appropriate in each case. The sample was previously coated with gold/platinum to give the sample the conductivity required for correct observation.

Fourier transform infrared spectroscopy (FTIR) was performed for the characterization of the starting materials (oils and PVA) as well as for the characterization of the nanofibers obtained. A JASCO FT/IR-4700 type A spectrophotometer (Jasco Spain, Madrid, Spain) with the ATR accessory (Jasco Spain, Madrid, Spain) was used. Sixteen spectra with a resolution of 4 cm^−1^ were performed for each sample. Thus, the main components of thyme, timol, and carvacol, as represented in Figure 1, were used to study FTIR spectra.

Differential scanning calorimetry (DSC) was performed with a Mettler-Toledo 821 DSC (Mettler-Toledo Inc., Schwerzenbach, Switzerland) at a heating rate of 10 °C min^−1^ in an oxygen atmosphere (60 mL min^–1^). The samples (approximately 3 mg in weight) were placed in a standard aluminum crucible with a volume of 40 μL and a sealing capacity to avoid the loss of material.

To obtain uniformity in the size distribution measurements, the image analysis software Image J 1.52p (Wayne Rasband, MD, USA) was used. Each of the images of the samples to be analyzed were suitably calibrated to obtain measurements in the correct units. The different measurements were transferred to Excel and the corresponding representations were obtained.

In order to determine whether the essential oil remained solid or liquid, a pressure test was applied. This test consists of applying a force of 5 N on the surface for 5 min and then analyzing the tested surface using scanning electron microscopy (SEM).

The antibacterial effect from the oils was tested according to ASTM E 2149-13 against Escherichia coli (Origin ATCC25922).

## 3. Results

### 3.1. Emulsion Characterization

The characteristics of the samples and the emulsions obtained were analyzed (see Table 1). Emulsions were based on 4% sage (S) or thyme (T) oil in an aqueous solution.

The correct characterization of the emulsions involves measuring the viscosity of each of the components as well as that of the prepared emulsion. Figure 1a shows the optical microscope image of a PVA emulsion (9% *w*/*v*) with the thyme essential oil (4%) made in a propeller homogenizer, working at 500 and 1000 rpm for 60 min. The manner in which the micro-drops of different sizes were obtained can be seen, demonstrating that, as expected given the hydrophilic and hydrophobic nature of the components, an emulsion was obtained between the PVA (aqueous) and the essential oil thyme (oil). The formation of aggregates or clusters of these droplets can also be seen at 500 rpm. After 15 min, the emulsion broke, easily separating the two phases and clearly showing the difference between the aqueous zone and the oil used, so this emulsion was discarded due to its low stability. When the revolutions were increased to 1000 rpm and the time was 60 min, the formation of smaller droplets was observed, as can be seen in Figure 2. It can also be seen (see area marked with a circle (Figure 2b)) that the emulsion was not stable, with some of the droplets tending to coalesce after 30 min, which is an indicator of low emulsion stability.

The poor stability of the emulsions together with the high dispersion of sizes seen when working with the propeller homogenizer and working between 500 and 1000 rpm led to the decision to work with a slot homogenizer at 7000 rpm for 3 min. In this case, it was observed that the emulsion was more stable. In the last 180 min after the end of the agitation, there was no tendency for them to melt, as in Figure 2b, therefore, this emulsion was stable for extrusion by means of the electrospinning process.

The emulsions were measured using Image J software, determining the size of the droplets created and their size distribution. Figure 3 shows that, when working at 7000 rpm, it is evident that the Sage droplets were uniformly distributed. The majority (76%) were between 0.75–1.5 µm, with a maximum peak (38%) of around 0.1–1 µm.

### 3.2. Nanofiber Characterization

#### 3.2.1. Organoleptic Test

The organoleptic test was carried out with 5 volunteers, and they were asked to identify the aroma, initially without reference and then offering a sample of the aroma of sage and pure thyme.

As can be seen from the results of the survey (Table 2), all respondents were able to identify an odor in the nanofiber samples that came from the emulsion with essential oils, while no odor was discernible in the samples with PVA alone. When asked to identify the perceived odor, the answers were different, including definitions such as grass (3) or mountain (1) or, more precisely, aromatic plants (1). In the case of essence identification, prior to being shown two essences (A = sage essential oil) (B = thyme essential oil), essence identification also occurred in 100% of these cases.

#### 3.2.2. SEM

The characterization of the nanofiber veils from the emulsions implies the need to distinguish two specific measurements. On the one hand, the PVA nanofibers were found in the same conditions as electrospinning without emulsion, which presented a tubular geometry and a cylindrical shape, as can be seen in Figure 4a. Moreover, the solvent evaporation was correct, and no porosity was observed in the fiber. On the other hand, we can distinguish the spheres that were generated due to the extrusion process in the emulsion of essential oil in the PVA solution (Figure 4b).

As mentioned previously, the extrusion of emulsions can result in the formation of core-shell nanofibers [15]. Çallıoğlu et al. [16] show that the addition of the essential oil in percentages of between 1% and 5% and surfactant help to eliminate the formation of what are called beds, which, in this work, we considered to be the formation of microspheres. One of the objectives of this work was to demonstrate whether they were solid (microspheres) or had the oil encapsulated (microcapsules).

When thyme essential oil was taken and extruded at different concentrations of oil (2% and 4%) and the same concentration of surfactant (1%), the formation of the spheres was maintained, so it can be understood that the formation of the spheres was attributable to the surface tension of the oil with the surfactant. In this case, given that this work focused on encapsulating the oil, 4% O/W formulation was considered to be optimal and the focus moved on to characterization.

Observational analysis of the nanofibers extruded with the 4% essential oil of sage and thyme showed the presence of spheres and not the presence of a central channel with the oil and the PVA polymeric membrane covering it, as would be expected in the formation of core-shell nanofibers. This effect, the appearance of spheres, was observed in the two essential oils tested, sage and thyme, as shown in Figure 5. This shows how an addition of 4% of essential oil generated the formation of spheres in the nanofibers, but also shows the differences in behavior between the two essential oils (a, b = sage vs. c, d = thyme) and the voltage applied to each of the samples during the electrospinning process (a, c = 14 kV vs. b, d = 20 kV).

The sage samples possessed a more regular surface, with a uniformly distributed nanofiber veil, while the thyme samples possessed clearly differentiated grooves, forming preferential directions in the deposition of the nanofibers on the collector. This effect was mitigated when we increased the potential difference from 14 kV to 20 kV (Figure 5c with respect to Figure 5d), such that the increase in voltage favored the homogeneity of the jet and it is likely that at 14 kV a jet that was not uniformly distributed was achieved. The thyme samples also showed a larger capsule size when the voltage was increased, behavior which was not observed in the sage samples. In order to be able to objectively analyze what was happening in each of the samples obtained, the particles were counted, and their size and distribution were measured using images at a higher magnification, specifically images at 5000 magnification (5 K).

Observation of the images at higher magnification (Figure 6) revealed two types of particles, some of which were embedded in the nanofiber, and others that seemed to have overcome the electrostatic forces and resulted in the formation of individual spherical particles trapped by the nanofiber network that was deposited previously or afterwards. These images show that a higher voltage (20 kV) created more spherical spheres, while at lower voltage, especially in the smaller particles, more elliptical shapes were obtained. An example of both is marked in red.

When the study was taken beyond visual observation and size measurements were taken for each of the spheres obtained, the analysis of the size distribution in the samples (Table 3) showed that, at higher field strength (20 kV), a greater homogeneity seemed to be obtained in the size distribution, this is due to the strong influence that existed in the particles of smaller sizes that were retained in the nanofibers with respect to the particles that managed to become independent of the jet and form themselves as independent spheres. In order to further the study, a particle analysis was carried out by segregating the results of the embedded particles from the ones that were not embedded.

Analysis of the size distributions (Table 3) demonstrated that the greatest homogeneity was in the distribution when working at 20 kV, both in the embedded and nonembedded microspheres, and for both sage and thyme essential oils. Thus, it can be affirmed that the greater intensity of the electric field resulted in more rounded microparticles with a more uniform size distribution. Furthermore, observing the contribution of the nonembedded spheres compared to the total number of spheres obtained in each of the tests carried out, the number of particles generated was higher for higher voltages, so it can be concluded that a greater number of independent particles were generated.

The nonembedded particles, which appeared as individual entities retained by the nanofiber network, were larger than the aligned ones (Table 4). This leads to the conclusion that the larger emulsion droplets, as they contain more oil, generated greater surface tension, a force which is capable of overcoming electrostatic forces, generating independent particles. The smaller particles were not able to break their bond with the jet polymer and were retained.

The microspheres from the sage emulsions resulted in particles with more homogeneous size distributions than those from the thyme. There seems to be a direct relationship between the sizes of the emulsions obtained and the microspheres obtained. The majority of droplets in the sage emulsion (0.75–1 µm) were slightly smaller in size than those in the thyme emulsion (1–1.25 µm). This relationship was maintained in the microspheres obtained, showing, for example, in the samples obtained at 20 kV, most of the sage spheres (38%) between 0.75 and 1 µm and thyme spheres (21%) between 1 and 1.25 µm. However, it is worth noting the presence of a large number of particles in the two sizes immediately above, which can be justified as an increase in the size (diameter) of the spheres as a consequence of the formation of a PVA membrane covering the essential oil.

The existence of independent particles requiring a polymer coating implied an increase in polymer consumption around the sphere and generated tensions in the polymer jet with the consequent stretching of the PVA, giving rise to much finer nanofibers, as can be seen in the measurements shown in the following table (Table 5).

During the microscopy analysis, some images were observed showing microspheres which had not been completely closed and some with perforations. Initially, it was assumed that there was not enough polymer to completely envelop the large spheres. However, a detailed analysis of the surface showed that some microspheres had very rough and porous surfaces characteristic of the rapid evaporation of the solvent [25]. It seems that evaporation of the highly volatile essential oil occurred (Figure 7) as a consequence of the generation of the Taylor cone and the evaporation of the solvent from the polymer as the nanofibers and microspheres reach the collector.

#### 3.2.3. FTIR

Fourier transform infrared spectroscopy (FTIR) was very useful for the identification of functional groups with special vibrations in certain areas of the spectrum between 4000–400 cm^−1^, however, a quantitative determination requires very precise calibrations, and is more complex to carry out due to the superposition of the vibration of the molecules in some areas of the spectrum [26]. This can cause fluctuations in the center of the band. In this case, we aimed to identify the presence of the essential oil in the nanofiber by means of FTIR, so as to be sufficient to identify the characteristic functional groups of the essential oil used and to determine the evolution of the PVA curve when the active compound was included. When the spectrum of the nanofibers obtained was analyzed and compared with the spectrum of the essential oil used, the presence of characteristic peaks of sage could be observed (Figure 8a). This figure shows the spectrum of polyvinyl alcohol in nanofiber on cotton fabric (blue), and the spectrum of the PVA nanofibers from an emulsion with sage.

These peaks were named A, B, C, and D. These are 4 characteristic peaks that have already been cited by some other authors who have analyzed sage [27].

The band centered at A corresponded to the peaks centered at 3300 cm^−1^ and essentially corresponded to OH stretching, a peak that was practically nonexistent in sage. However, next to this band we found band B, which corresponded to the band 2956–2849 cm^−1^, assigned to both symmetrical and asymmetrical CH stretching (CH_3_ and CH_2_) [27] and which, when tested with the essential oil sage, also presented considerable intensity. Thus, the presence of sage can be identified as an increase in the CH band with respect to the OH band. This can be seen in Table 6 where the spectrum of the electrospun PVA nanofibers on cotton (PVA), the spectrum of the sage essential oil (S) as supplied by the supplier, and the spectrum of the electrospun PVA-sage emulsion are displayed. The intensity analysis showed that the presence of sage caused the CH stretching band intensity (I_2915_) to increase with respect to the OH stretching (I_3300_). Therefore, when we analyzed the ratio of these two bands (I_3300_ /I_2915_), there was a large difference between PVA (1.8443) and sage (0.0481). This means that the presence of sage in the PVA was demonstrated by a decrease in this ratio with respect to the PVA without sage.

However, camphor and thujone, characteristic of sage, presented C=O stretching vibrations centered around 1730 cm^−1^. In addition, there was the presence of pinene, evidenced by the -C=C- alkene bond, the band of which was centered at 1640 cm^−1^ [16,27]. These bands are characteristic of the terpenes found in essential oils [28]. If analyzed in the spectra of nanofibers (Figure 8) these peaks corresponded to the named C and D bands, respectively, of the sage spectrum. When we analyzed the behavior of these bands in the sage sample used in the study, it was observed that the presence of pinene was much lower than that of camphor and thujone, showing a band with a very characteristic peak at 1730 cm^−1^. Table 6 shows that PVA nanofibers with sage decreased the ratio (I_1640_/I_1734_ = 1.3290) compared to that of PVA without sage (I_1640_/I_1734_ = 1.7189). Therefore, these band intensities also confirmed the presence of sage inside the nanofibers.

When we move to the sage and thyme essential oil, the bibliography indicates that it is mainly composed of the volatile agents: thymol and terpinene [23], although other authors indicate that its main composition is thymol and carvacrol [28], which are very similar in their chemical structure, as can be seen in Figure 1. However, other authors assign the main component as carvacrol [16].

The peak at 3001 cm^−1^ was associated with the CH stretching vibration in the benzene ring of the thymol [26] and was also present in the carvacrol, a band identified by the letter F, according to the spectrum shown in Figure 8b. This peak is characterized by being more intense than the OH stretching band centered at 3300 cm^−1^, labelled with the letter E. The ratio between these bands, I_3300_/I_3001_, was very different between thyme (where I_3300_/I_3001_ < 1) and PVA (where I_3300_/I_3001_ > 1), so the presence of thyme essential oil was observed with the decrease in this ratio as can be seen in Table 7. This is due to the fact that the E band, centered around 3300 cm^−1^, which was practically constant, was slightly increased in the case of thyme essential oil due to the presence of NH groups, whose band is found in this region [16]. There were different absorption bands related to amide I (C=O stretching), amide II (NH bending and CH stretching), and amide III (CN stretching plus NH phase bending). Amide I, amide II, and amide III peaks appeared in the spectrum around 1705 cm^−1^, 1516 cm^−1^, and 1232 cm^−1^, respectively. In the case that they appear in thyme essential oil, this may be due to the presence of lignin residues that have been included in the essential oil even after purification treatments, as shown by Choi et al. [29]. Table 7 and Table 8 show a summary of the values obtained in each of the characteristic bands of the study for the electrospun thyme essential oil.

The peak around 800–820 cm^−1^ was also associated with the presence of thymol and carvacrol [19,28] as a consequence of the presence of substituted aromatic rings, a band which was clearly seen in the spectrum of the thyme essential oil analyzed, which we labelled I, and which was also identified by Popa et al. [26]. When the ratio I_812_/I_3300_ was established, it was observed that I_812_/I_3300_ < 1 for PVA (blue) and I_812_/I_3300_ > 1 for thyme essential oil (grey). The spectrum of the PVA nanofibers with essential oil (orange) gave values of I_812_/I_330_ > 1, so this ratio also demonstrated the presence of thyme essential oil in the PVA nanofibers.

Observation of the spectrum showed two strong bands in the essential oil of Thyme, but which did not create such a strong influence as in the two previously analyzed bands, the band at 1420 cm^−1^ (labeled H) and the band centered at 1226 cm^−1^ (labeled I), the results of which can be seen in Table 8.

The H band (1420 cm^−1^) was also characteristic of thyme and was attributed to CO stretching vibrations (amide), CC stretching from phenyl groups, COO-symmetric stretching, and CH_2_ bending [28]. In PVA, only the CH_2_ bending vibration was seen, so the spectrum of the essential oil of thyme (green) offered greater intensity in the G band than PVA (blue) due to the greater number of groups vibrating at this wavelength. When the essential oil was incorporated into the PVA nanofibers, this band was also slightly increased.

The band at 1226 cm^−1^, assigned to amide group III, also showed the presence of thyme essential oil in the nanofibers obtained by increasing the ratio I_1226_/I_3300_ when thyme was introduced into the PVA, and indicated not only the presence of the essential oil, but also that the essential oil included traces of lignin from the plant [29]. When we analyzed the G band assigned to the presence of the terpenes characteristic of essential oils [28], it was observed that the I_1734_/I_1640_ ratio detected was not as pronounced as in the case of sage, because the 1734 cm^−1^ band did not appear as such but a band was found around 1700 cm^−1^, which corroborates the presence of lignin, so in this case it cannot be considered an appropriate indicator to determine the presence of thyme. Although there was evidence of an increase in the PVA electrospun with the emulsion of this band with respect to the PVA, this demonstrates the presence of the essential oil containing lignin.

#### 3.2.4. DSC

The differential scanning calorimetry (DSC) technique allows the stability of materials versus temperature to be categorized. In this case, the PVA nanofibers without essential oil, the essential oil, and the nanofibers obtained from the essential oil with the microcapsules were tested. The following figures show the behavior of the different samples tested.

As can be observed from the thermograms (Figure 9a,b), both sage and thyme essential oils showed characteristic curves with degradation starting at around 180 °C, and in the case of sage, degradation was also seen at around 70 °C.

The PVA did not show any alterations in these areas, but showed an abrupt peak at around 210 °C. When the nanofibers included the essential oils, an alteration of the curve was seen because of the influence of the essential oil, which showed that the microspheres contained essential oil, as expected from the results of the organoleptic test and in view of the FTIR results. However, it could not be determined whether the oil had been solidified to create solid microspheres.

#### 3.2.5. Pressure Test

To determine whether the essential oil seen in the FTIR spectrum was encapsulated in the spheres or whether it had partially solidified and evaporated, a pressure test was carried out on the nanofiber veils. This test consisted of applying a force of 5 N on the surface for 5 min and then analyzing the tested surface by microscopy (SEM) (Figure 10).

Some of the microspheres degraded as a consequence of the pressure exerted on them, and a sort of polymeric mass from the broken spheres was observed. They lost their spherical geometry and became part of a flat layer together with the nanofibers. In some cases (Figure 10a), some partially deflated capsules were also seen. It was also observed that it was essentially the larger ones that degraded due to the pressure, and some of the smaller ones remained undegraded. This effect has already been observed in the application of microcapsules on textiles [30], where results showed that the smaller microcapsules fit between the fibers, finding a gap that frees them from the pressure, and higher pressures are required to ensure enough compaction to cause degradation. In the case of the test on thyme samples (Figure 10b), the appearance of a flattened polymeric membrane was detected, without detecting the presence of empty capsules, the obvious conclusion being that the test contributed to the complete rupture of the membrane, releasing the oil.

#### 3.2.6. Antibacterial Effect

The antibacterial effect of both oils was tested, and its high efficiency can be observed in Table 9. Once the oil was encapsulated into the PVA and crosslinked by the nanofiber mesh, the antibacterial effect remained for both the sage and thyme oils. These results show that encapsulation does not impact the antibacterial effect, which may be due to the presence of a semipermeable shell.

## 4. Discussion

In order to carry out a correct extrusion, it is important to start with stable, homogeneous emulsions. If the two emulsions from different essential oils were compared, as expected, for the same composition, at higher revolutions, a smaller sphere size was obtained. There was also a difference in behavior between sage and thyme, with the thyme droplets being slightly larger than those of the sage, which coincides with the higher viscosity of the sage emulsion.

The identification of aroma in the samples that came from emulsions with sage and thyme essential oil, as well as the correct identification of the aroma that had been encapsulated, suggests that the nanofiber veils created have led to the retention of the essential oil in some way.

The formulation of 1% surfactant with 4% oil that was used was sufficient to achieve uniform emulsions which result in microspheres either embedded in the nanofiber or as an individual entity which was interlinked by the multiple nanofibers generated in the process.

A higher voltage in the electrospinning process resulted in more regular microspheres, regular distributions in the nanofiber veil of both the nanofibers themselves and the generated spheres, as well as larger microspheres.

The generated microspheres were closely related to the starting emulsion droplet sizes, although an increase in the % of microspheres was observed at slightly larger sizes and was attributed to the formation of a PVA polymeric membrane around the oil drop in the emulsion.

The components of the essential oils used (sage and thyme) in the PVA nanofibers can be identified in PVA nanofiber by FTIR. Specifically, the results showed that the bands corresponding to the OH stretching vibrations centered at 3300 cm^−1^ and CH stretching centered around 3000 cm^−1^ for thyme and 2915 cm^−1^ for sage were decisive for the two oils used. In addition, each of the oils used showed characteristic bands that demonstrated alterations in the behavior of the PVA.

Calorimetric tests confirmed the conclusions reached using FTIR: the presence of essential oils trapped inside the electrospun spheres.

The pressure of 5 N on the microspheres caused many of the larger ones to lose their spherical shape as a consequence of having been subjected to the pressure, and therefore the oil that was encapsulated inside them was released. Thus, it can be affirmed that the spheres that were observed were in fact microcapsules containing the essential oil and not microspheres.

The presence of the essential oils in liquid state, as demonstrated by the pressure test, was confirmed. It was also concluded that the detail of some of the microspheres that had apparently lost the essential oil by evaporation through the membrane during the encapsulation process corresponded to smaller spheres.

## 5. Conclusions

The organoleptic analysis revealed the presence of the aromas in the nanofiber veils and, therefore, we proceeded with the analysis of the samples obtained in order to identify whether the aroma was retained in the fiber and whether it had solidified or was retained in the form of oil.

SEM made it possible to observe the geometry of PVA nanofibers electrospun from emulsions with the essential oils sage and thyme, showing that there is a certain emulsion size that is capable of generating tensions, and thus deformations, in the nanofibers. When the size of the emulsion is large enough, the tensions are such that the nanofibers break, protecting the essential oil in the form of a nanosphere. The field strength influences the size of the spheres and the homogeneity of the spheres in terms of size distribution.

The size ratio of the droplets in the emulsion and the microspheres in the nanofiber veils suggests that an increase in the size of the microspheres is generated as a consequence of the formation of a polymeric membrane covering the emulsion droplets. The formation of this polymeric membrane around the oil droplet leads to a consumption of PVA and the consequent tension of the jet, resulting in PVA nanofibers with diameters considerably smaller than those obtained when starting with PVA without oil.

Given the FTIR results obtained, it can be concluded that the FTIR spectrum using the ATR technique is capable of detecting the presence of the essential oils used to obtain the emulsions. This leads to the following conclusion: that both sage oil and thyme oil had been trapped in the nanofibers and the electrostatic field generated for the evaporation of the PVA solvent was not powerful enough to evaporate all the volatile compounds such as the thymol present in the thyme oil or those of sage, such that these were retained and could therefore confer some of their properties on to the nanofibers.

The DSC calorimetry test corroborated the presence of oil in the nanofiber veils, confirming that the microspheres were in fact essential oil microcapsules.

The pressure test on the nanofiber veils revealed that the microspheres observed in the veils were spherical cavities inside which part of the emulsified essential oil was housed, so it can be concluded that the microspheres were in fact microcapsules.

Therefore, the results show that in the formation of the Taylor cone, the essential oil is retained by the PVA polymer, maintaining its liquid state, thus proving the hypothesis stated at the beginning of the paper.

## Figures and Tables

**Figure 1 polymers-14-05242-f001:**
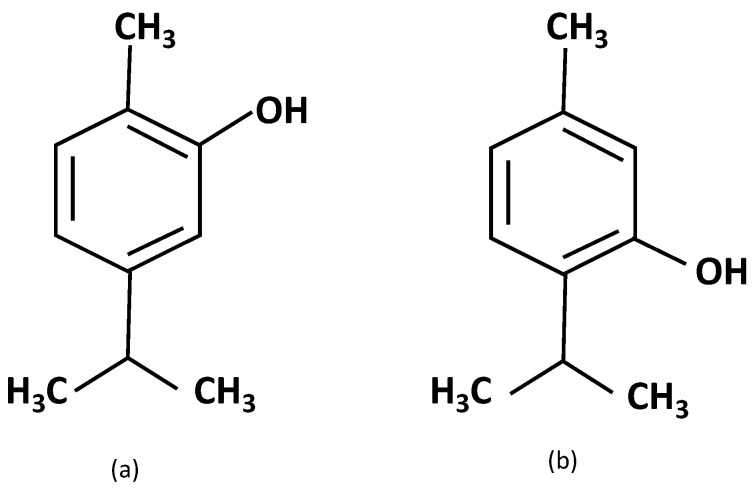
Structure of thymol and carvacrol present in thyme essential oil. (**a**) Carvacol; (**b**) thymol.

**Figure 2 polymers-14-05242-f002:**
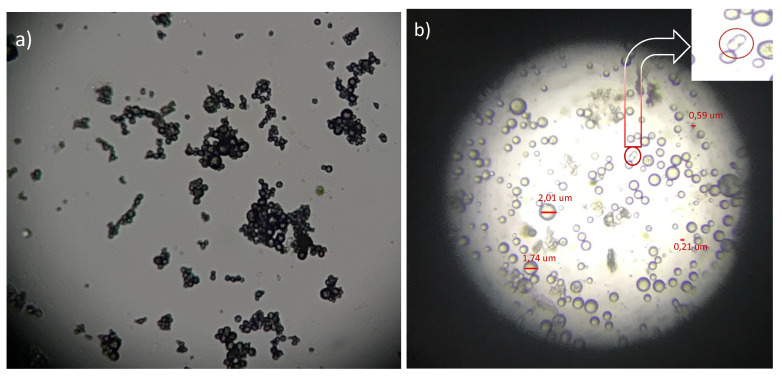
PVA emulsion from thyme essential oil under optical microscope. (**a**) 500 rpm; (**b**) 1000 rpm.

**Figure 3 polymers-14-05242-f003:**
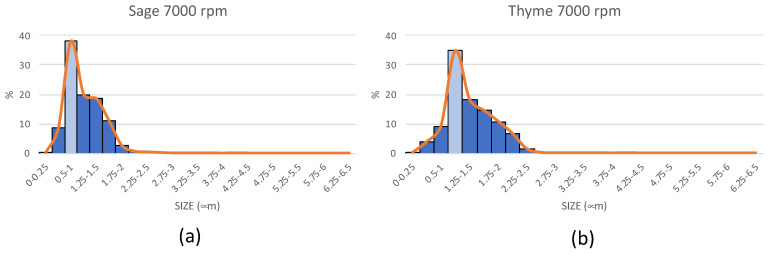
Size distribution of sage and thyme 4% emulsions on PVA solution. (**a**) Sage; (**b**) thyme.

**Figure 4 polymers-14-05242-f004:**
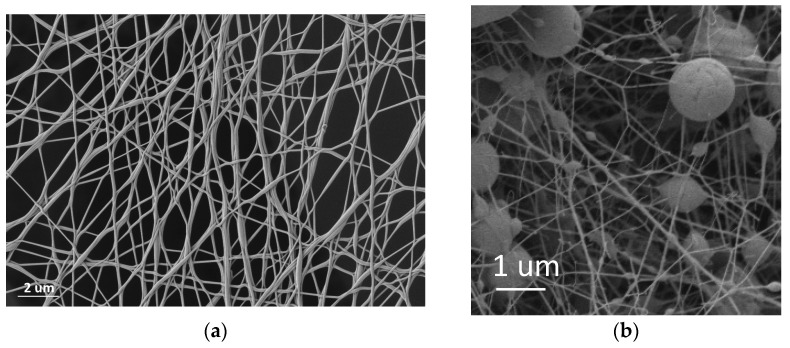
PVA nanofiber veil. (**a**) PVA 5 K; (**b**) PVA + 4% sage essential oil 5 K.

**Figure 5 polymers-14-05242-f005:**
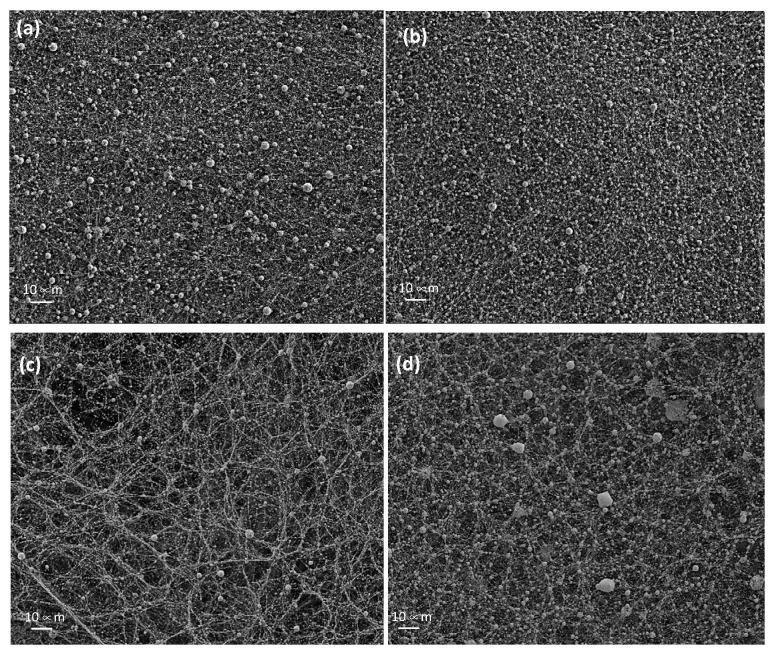
Nanofibers from essential oil emulsions. (**a**) Sage 14 kV; (**b**) sage 20 kV; (**c**) thyme 14 kV; (**d**) thyme 20 kV.

**Figure 6 polymers-14-05242-f006:**
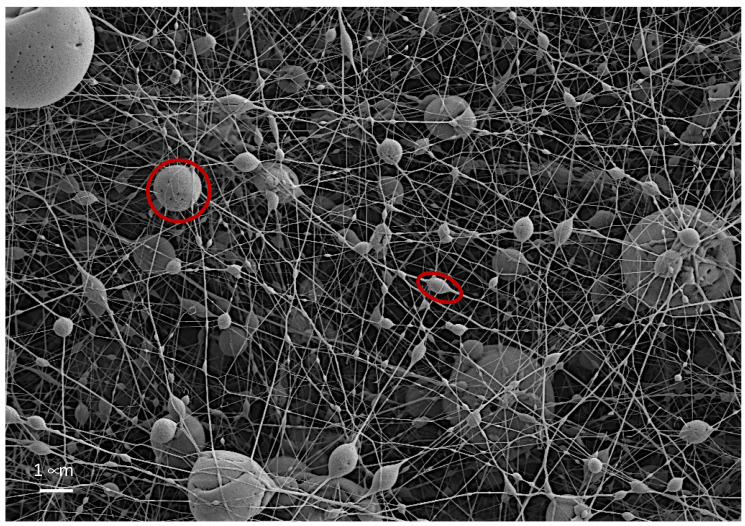
Differentiation between the particles formed in the nanofiber web electrospun with essential oil (thyme 14 kV).

**Figure 7 polymers-14-05242-f007:**
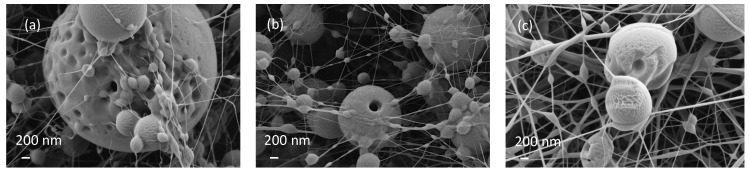
Shell detail from some microspheres.

**Figure 8 polymers-14-05242-f008:**
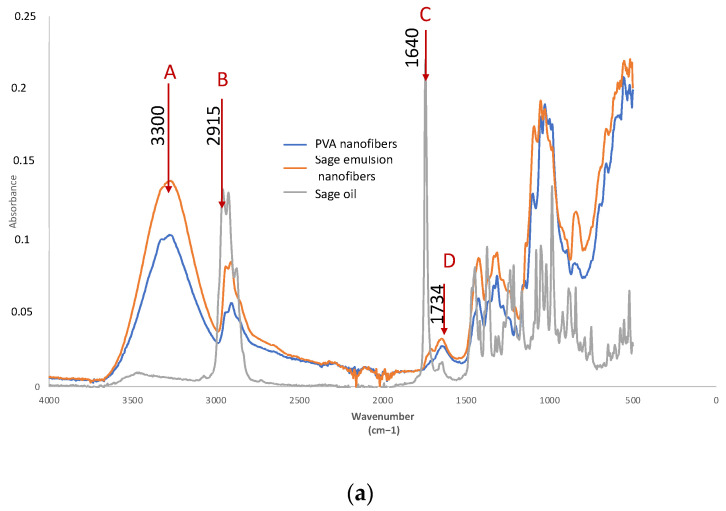
Infrared spectrum of nanofibers. (**a**) Sage essential oil; blue line: PVA nanofibers; orange line: sage emulsion nanofibers; grey line: sage oil. (**b**) Thyme essential oil; blue line: PVA nanofibers; orange line: thyme emulsion nanofibers; grey line: thyme oil.

**Figure 9 polymers-14-05242-f009:**
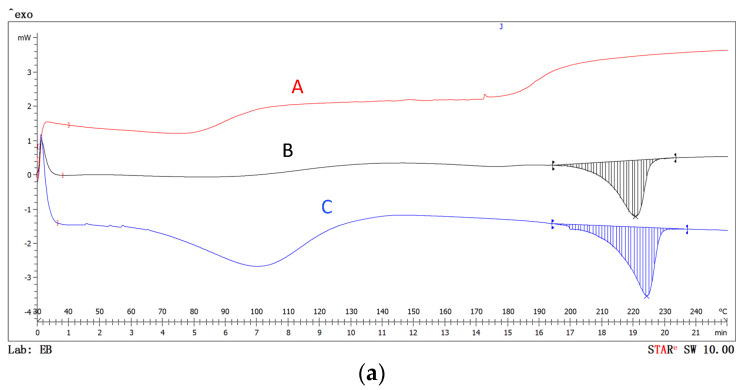
DSC of nanofibers. (**a**) Sage: (A) sage essential oil; (B) PVA nanofibers; (C) PVA−sage nanofibers. (**b**) Thyme (A) thyme essential oil; (B) PVA nanofibers; (C) PVA−thyme nanofibers.

**Figure 10 polymers-14-05242-f010:**
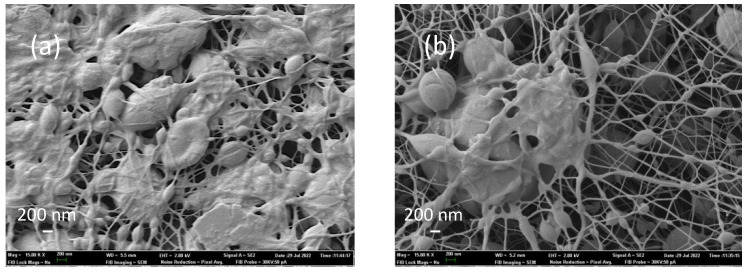
Surface of the nanofiber veils with beads after the pressure test. (**a**) Sage; (**b**) thyme.

**Table 1 polymers-14-05242-t001:** Properties of the emulsions used.

Reference	PVA 9%	S	T	4% S	4% T
Viscosity (cP)	173.2	30.7	28.9	338	230
Conductivity (μS)	270	--	--	242	268
Surface tension (mN/m)	70	27.82	32.76	37.12	33.25

**Table 2 polymers-14-05242-t002:** Organoleptic test results.

Reference	PVA 9%	PVA + S 14 kV	PVA + S 20 kV	PVA + T 14 kV	PVA + T 20 kV
Odor identification (volunteers)	No (5)	Yes (5)	Yes (5)	Yes (5)	Yes (5)
Fragrance identification (volunteers)	-	Yes (5)	Yes (5)	Yes (5)	Yes (5)

**Table 3 polymers-14-05242-t003:** Size distribution of microspheres from sage and thyme microemulsions electrospun at 14 kV or 20 kV.

Size (μm)	S 14 kV	S 20 kV	T 14 kV	T 20 kV
0–0.25	18.02	15.23	25.5	41.83
0.25–0.5	29.73	43.71	52.5	31.73
0.5–0.75	18.92	17.22	12	3.85
0.75–1	15.32	9.27	5	3.37
1–1.25	5.41	7.28	1.5	5.29
1.25–1.5	8.11	3.97	0.5	4.33
1.5–1.75	0.00	0.66	1	3.37
1.75–2	0.90	1.32	0.5	1.44
2–2.25	0.00	0.66	0.5	3.85
2.25–2.5	0.00	0.00	0.5	0.48
2.5–2.75	0.90	0.66	0	0.48
2.75–3	0.00	0.00	0	0.00
3–3.25	0.00	0.00	0	0.00
3.25–3.5	1.80	0.00	0	0.00
3.5–3.75	0.90	0.00	0	0.00

**Table 4 polymers-14-05242-t004:** Comparison between embedded and nonembedded spheres.

Size (μM)	S 14 kV %		S 20 kV %		T 14 kV %		T 20 kV %	
	No Emb.	Emb.	No Emb.	Emb.	No Emb.	Emb.	No Emb.	Emb.
0–0.25	0.00	20.62	0.00	18.85	0.00	28.18	0.00	55.41
0.25–0.5	0.00	34.02	0.00	54.10	0.00	58.01	0.00	42.04
0.5–0.75	0.00	21.65	6.90	19.67	31.58	9.94	9.80	1.91
0.75–1	14.29	15.46	37.93	2.46	21.05	3.31	11.76	0.64
1–1.25	14.29	4.12	27.59	2.46	10.53	0.55	21.57	0.00
1.25–1.5	35.71	4.12	17.24	0.82	5.26	0.00	17.65	0.00
1.5–1.75	0.00	0.00	3.45	0.00	10.53	0.00	13.73	0.00
1.75–2	7.14	0.00	3.45	0.82	5.26	0.00	5.88	0.00
2–2.25	0.00	0.00	0.00	0.82	5.26	0.00	15.69	0.00
2.25–2.5	0.00	0.00	0.00	0.00	5.26	0.00	1.96	0.00
2.5–2.75	7.14	0.00	3.45	0.00	0.00	0.00	1.96	0.00
2.75–3	0.00	0.00	0.00	0.00	0.00	0.00	0.00	0.00
3–3.25	0.00	0.00	0.00	0.00	0.00	0.00	0.00	0.00
3.25–3.5	14.29	0.00	0.00	0.00	0.00	0.00	0.00	0.00
3.5–3.75	7.14	0.00	0.00	0.00	0.00	0.00	0.00	0.00

**Table 5 polymers-14-05242-t005:** Nanofiber cross-section dimensions (nm).

Reference	PVA 9%	PVA + S 14 kV	PVA + S 20 kV	PVA + T 14 kV	PVA + T 20 kV
Size (nm)	0.2256	0.098	0.0972	0.1408	0.1238

**Table 6 polymers-14-05242-t006:** Band intensity of FTIR spectra of PVA, sage, and electrospun PVA−sage nanofiber veils.

Reference	A	B	A/B	C	D	C/D
	I_3300_	I_2915_	I_3300_/I_2915_	I _1640_	I_1734_	I_1640_/I_1734_
PVA	0.1007	0.0546	1.8443	0.0263	0.0153	1.7189
SAGE (S)	0.0062	0.1288	0.0481	0.0156	0.219	0.0712
PVA-S Electrospun	0.1381	0.0827	1.6699	0.0307	0.0231	1.3290

**Table 7 polymers-14-05242-t007:** Band intensity of FTIR spectra of PVA, thyme, and electrospun PVA−thyme nanofiber veils.

Reference	E	F	E/F	I	I/E
	I_3300_	I_2915_	I_3300_/I_2915_	I_812_	I_812_/I_3300_
PVA	0.1018	0.0532	1.9135	0.0805	0.7908
Thyme (T)	0.1100	0.1261	0.8723	0.1329	1.2082
PVA-T Electrospun	0.0963	0.0639	1.5070	0.1011	1.0498

**Table 8 polymers-14-05242-t008:** Intensity of other characteristic bands of the FTIR spectrum of the less-relevant thyme-essential-oil nanofiber veils.

Reference	E	H	H/E	I	I/E	G1	G2	G1/G2
	I_3300_	I_1420_	I_1420_/I_3300_	I_1226_	I_1226_/I_3300_	I_1700_	I_1640_	I_1700_/I_1640_
PVA	0.1018	0.0535	1.1863	0.0451	0.4430	0.0123	0.0265	0.4642
Thyme (T)	0.1100	0.1112	1.3745	0.0809	0.7355	0.032	0.0491	0.6517
PVA-T Electrospun	0.0963	0.068	1.2274	0.0554	0.5753	0.0156	0.0188	0.8298

**Table 9 polymers-14-05242-t009:** Intensity of other characteristic bands of the FTIR spectrum of the less-relevant thyme-essential-oil nanofiber veils.

% Reduction (cfu/mL)	S	T	S 20 kV	T 20 kV
	Oil	Oil	Nanofibers	Nanofibers
Escherichia coli	99.99	99.99	99.99	99.99

## Data Availability

The data presented in this study are available in Microcapsules. *Polymers*
**2022**, *14*, 5242. https://doi.org/10.3390/polym14235242.

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
