# Peer review of "Liquid Oil Trapped inside PVA Electrospun Microcapsules"

_polymers, 2022, doi:10.3390/polym14235242_

Round 1
Reviewer 1 Report
The aim of this study was the encapsulation of two essential oils sage and thyme in the polyvinyl alcohol (PVA) matrix using electrospinning method. Electrospinning process was directed in order to get microcapsules of oil inside a net of nanofibers. The electrospun veil was characterized by organoleptic test, SEM, FTIR, DSC, and pressure test. The results confirmed presence of essential oils in liquid form trapped inside the electrospun spheres. The good correlation between emulsion size and the final microcapsules is noticed. The nanofibers diameter was considerable reduced in comparison to prepared nanofibers without the oil.
The results presented in this research are interesting, but the English is poor. The message to the reader needs to be clearly presented and correlated through whole manuscript text. The authors need to address the issues listed below before re-submission of the paper:
Major issues:
1. The authors suggested that thyme and sage essential oils are chosen due their good antioxidant and antimicrobial properties but neither one of these tests are shown in the manuscript. The authors should perform antioxidant and/or some antibacterial tests to show the difference in the biocide efficacy of PVA nanofibers before and after encapsulation of essential oils.
2. The last paragraph in introduction section needs to be amended to be in correlation with the aim of this research. Aim of the research, novelty aspect and conclusion marks are not well-correlated.
3. The charge of the deposited fibers has influence of the morphology of the fibers but there are other parameters such as solvent, which also contribute. The authors should make some comments related to the solvent factor on morphology of electrospun fibers.
Minor issues:
1. FTIR spectra are not clearly presented. The major peaks need to be labelled with numbers commented in the manuscript text.
2. DSC Figures need to be properly labelled and clearly presented. The text within Figures 11 and 12 are not visible and should be excluded. The DSC curves need to be properly labelled e.g. a, b, and c or similar.
Author Response
Authors would like to acknowledge your time devoted to the revision of our manuscript and we really appreciate your comments as they will improve our final result. You can find below our answers.
The aim of this study was the encapsulation of two essential oils sage and thyme in the polyvinyl alcohol (PVA) matrix using electrospinning method. Electrospinning process was directed in order to get microcapsules of oil inside a net of nanofibers. The electrospun veil was characterized by organoleptic test, SEM, FTIR, DSC, and pressure test. The results confirmed presence of essential oils in liquid form trapped inside the electrospun spheres. The good correlation between emulsion size and the final microcapsules is noticed. The nanofibers diameter was considerable reduced in comparison to prepared nanofibers without the oil.
The results presented in this research are interesting, but the English is poor. The message to the reader needs to be clearly presented and correlated through whole manuscript text. The authors need to address the issues listed below before re-submission of the paper:
Many thanks for the effort of reading and reviewing the paper. Your valuable comments can help us to improve the paper. Regarding the language, we have had the paper reviewed by an English native man.
Major issues:
- The authors suggested that thyme and sage essential oils are chosen due their good antioxidant and antimicrobial properties but neither one of these tests are shown in the manuscript. The authors should perform antioxidant and/or some antibacterial tests to show the difference in the biocide efficacy of PVA nanofibers before and after encapsulation of essential oils.
Although the biocide (antibacterial and virucide) test is the focus of a different paper, we agree with you, we don´t demonstrate the encapsulated spheres show this activity and have included some results in the text.
Section 2: Materilas & Methods
The antibacterial effect form the oils was tested according to ASTM E 2149-13 against Escherichia coli (Origin ATCC25922).
Section 3: Results
The antibacterial effect of both oils was tested and its high efficiency can be observed in table 9. Once the oil is encapsulated into the PVA and crosslinked by the nanofibers net, the antibacterial effect remains for both sage and thyme oil. These results show the encapsulation does not affect the antibacterial effect what can be due to the presence of semipermeable shell.
Table 9. Intensity of other characteristic bands of the FTIR spectrum of the less relevant Thyme essential oil nanofiber veils
% Reduction (cfu/mL) |
S |
T |
S 20 kV |
T 20kV |
oil |
oil |
nanofibers |
nanofibers |
|
Escherichia Coli |
99,99 |
99,99 |
99,99 |
99,99 |
- The last paragraph in introduction section needs to be amended to be in correlation with the aim of this research. Aim of the research, novelty aspect and conclusion marks are not well-correlated.
All right, we agree with the reviewer this paragraph is incomplete. Thus, we added what you can find below.
The objective of this paper is to demonstrate the possibility of generating PVA microencapsulated oil from emulsion electrospinning. Authors pretend to demonstrate the presence of the oil within the PVA capsules and characterize the capsules shape and size, determine the electrospun conditions and test whether the encapsulation keeps antibacterial properties from oil.
- The charge of the deposited fibers has influence of the morphology of the fibers but there are other parameters such as solvent, which also contribute. The authors should make some comments related to the solvent factor on morphology of electrospun fibers.
We have included the comment you can find below.
The characterization of the nanofiber veils from the emulsions implies the need to distinguish two specific measurements. On the one hand, we find the PVA nanofibers in the same conditions of electrospinning without emulsion, which present a tubular geometry, and a cylindrical shape, as can be seen in the following image (Figure 3a) furthermore, we can confirm the solvent evaporation is correct and no porosity can be observed in the fiber.
Minor issues:
- FTIR spectra are not clearly presented. The major peaks need to be labelled with numbers commented in the manuscript text.
It has been redesigned including pick numbers for the main bands. Figure captions have also been completed as follows.
Figure 8. Infrared spectrum of nanofibers. a) with Sage essential oil. Blue line: PVA nanofibers; orange line: sage emulsion nanofibers; grey line: sage oil. b) Thyme essential oil. Blue line: PVA nanofibers; orange line: thyme emulsion nanofibers; grey line: thyme oil.
Figure 9. DSC of nanofibers. a) Sage: A) Sage essential oil; B) PVA nanofibers; C) PVA-Sage nanofibers. b) Thyme A) Thyme essential oil; B) PVA nanofibers; C) PVA-Thyme nanofibers.
- DSC Figures need to be properly labelled and clearly presented. The text within Figures 11 and 12 are not visible and should be excluded. The DSC curves need to be properly labelled e.g. a, b, and c or similar.
It has been redesigned including labeling for each curve and figures caption description have been completed.
Figure 11. DSC of Sage nanofibers. A) Sage essential oil; B) PVA nanofibers; C) PVA-Sage nanofibers.
Figure 12. DSC of Thyme nanofibers. A) Thyme essential oil; B) PVA nanofibers; C) PVA-Thyme nanofibers.
Reviewer 2 Report
this manuscript provided detail information about electrospinning fiber and properties.
it could be publsihed after revision
1. the title need modify, it is not attractive
2. the introduction part is too short, more related information should be added
3. there are a lot of related papers published, what is the major different of this manuscript with others?
4. what si the boiling point of Sage (S) and Thyme (T)? it related with the evoperation
5. only FTIR is not enough to identify the chemical structure of thr fiber
6. what is the stablility of the Sage (S) and Thyme (T) inside the fibers?how to make sure it is inside the fiber?
7. there are too many figure to show the fiber size, it could be put together in a table
8. from figure 7, we can see there are a lot of porous on the fibers, what is inside?
9. figure 9 could put in experimental part
10. there are too many figures, soem of them could be put together
11. the discussion part is too short, on results part, there are some discussion on the other hands
12. conclusion part need modify
13. the English need improvement
14. the references should keep the same format
Author Response
Authors would like to acknowledge your time devoted to the revision of our manuscript and we really appreciate your comments as they will improve our final result. You can find below our answers.
this manuscript provided detail information about electrospinning fiber and properties.
it could be publsihed after revision
- the title need modify, it is not attractive
We suggest a new tittle:
Polyvinyl alcohol to create nanofibers net with microcapsules
Liquid oil trapped inside PVA electrospun microcapsules.
Although another alternative could be:
Essential oils microcapsules by PVA emulsion electrospinning.
- the introduction part is too short, more related information should be added
We have included three new paragraphs which we consider can be significative content for the reader to understand.
The production of nanofibers through electrospinning is controlled by a multitude of parameters which affect the final result, these are solution parameters such as: polymer concentration, viscosity, surface tension, molecular weight, conductivity, and solvent volatility, variables of the electrospinning process: voltage, supply flow, type of collecting surface and distance between electrodes and environmental parameters: humidity, temperature and air pressure.
…..
The encapsulation of active principles, essential oils, drugs, enzymes, vitamins, etc., inside the nanofibers has meant a great advance in different industrial sectors, such as its use in filtration, in the controlled release of drugs, in dressings for wounds or in the immobilization of enzymes. For this purpose there is the possibility to electrospun the active compound dispersed within the polymer, or an emulsion.
…..
The encapsulation of active principles, essential oils, drugs, enzymes, vitamins, etc., inside the nanofibers has meant a great advance in different industrial sectors, such as its use in filtration, in the controlled release of drugs, in dressings for wounds or in the immobilization of enzymes. For this purpose there is the possibility to electrospun the active compound dispersed within the polymer, or an emulsion.
- there are a lot of related papers published, what is the major different of this manuscript with others?
We demonstrate we can encapsulate the oil and in te revised version we also demonstrate the antibacterial effect remains. The antibacterial test has been included in the revised form. The antibacterial effect form the oils was tested according to ASTM E 2149-13 against Escherichia coli (Origin ATCC25922).
Section 2: Materilas & Methods
The antibacterial effect of both oils was tested and its high efficiency can be observed in table 9. Once the oil is encapsulated into the PVA and crosslinked by the nanofibers net, the antibacterial effect remains for both sage and thyme oil. These results show the encapsulation does not affect the antibacterial effect what can be due to the presence of semipermeable shell.
Table 9. Intensity of other characteristic bands of the FTIR spectrum of the less relevant Thyme essential oil nanofiber veils
% Reduction (cfu/mL) |
S |
T |
S 20 kV |
T 20kV |
oil |
oil |
nanofibers |
nanofibers |
|
Escherichia Coli |
99,99 |
99,99 |
99,99 |
99,99 |
4. what si the boiling point of Sage (S) and Thyme (T)? it related with the evoperation.
Sage evaporation temperature ois 210º C and Thyme is 195ºC. This confirms the degradation on DSC.
- only FTIR is not enough to identify the chemical structure of thr fiber
Authors, completely agree with you, but our purpose is not to determine the chemical structure of the fibre but the presence of essential oil. This is the reason why we also included the DSC test, and currently in the revision file you can find the antibacterial effect as well.
- what is the stablility of the Sage (S) and Thyme (T) inside the fibers?how to make sure it is inside the fiber?
Sage and Thyme are not inside the fiber as they narrow considerably, they are inside the capsules. This is the reason why we conduct the pressure test to demonstrate the spheres are deflated when the oil is spread around.
- there are too many figure to show the fiber size, it could be put together in a table
We have converted figure 5 and figure 6 into table 3 and table 4 as your suggestion. Nevertheless, authors consider graphs show the reader the tendency faster than a table which requires more exhaustive analysis.
- from figure 7, we can see there are a lot of porous on the fibers, what is inside?
Considering the empty spheres shown in figure 10 (from the new version) are deflated, we consider there is still oil inside but we cannot demonstrate it.
- figure 9 could put in experimental part
It has been moved, so that now it is Figure 1.
- there are too many figures, some of them could be put together
We organized figures and presented both figures for DTIR and both for DSC together.
- the discussion part is too short, on results part, there are some discussion on the other hands
Authors completely agree with the reviewer so taking as an example some published paper we have merged Results and discussion into one unique section.
- conclusion part need modify
They have been modified, authors hope we could understand the reviewer purpose.
- the English need improvement
The document has been reviewed by a native English person.
- the references should keep the same format
They have been carefully revised.
Round 2
Reviewer 1 Report
The authors the reviewer's comments in the revised manuscript. The amended manuscript is acceptable for publication.
Reviewer 2 Report
accept